# Privacy Protection in Personalized Diffusion Models via Targeted Cross-Attention Adversarial Attack

**Xide Xu**[1,2,*] **Muhammad Atif Butt**[1,2]**, Sandesh Kamath**[1]**, Bogdan Raducanu** [1,2]
[1]Computer Vision Center
[2] Universitat Autònoma de Barcelona
{xide, mabutt, skamath, bogdan}@cvc.uab.es

## Abstract

The growing demand for customized visual content has led to the rise of personalized text-to-image (T2I) diffusion models. Despite their remarkable potential, they pose significant privacy risk when misused for malicious purposes. In this paper, we propose a novel and efficient adversarial attack method, **Co**ncept **P**rotection by **S**elective **A**ttention **M**anipulation (*CoPSAM*) which targets only the cross-attention layers of a T2I diffusion model. For this purpose, we carefully construct an imperceptible noise to be added to clean samples to get their adversarial counterparts. This is obtained during the fine-tuning process by maximizing the discrepancy between the corresponding cross-attention maps of the user-specific token and the class-specific token, respectively. Experimental validation on a subset of CelebA-HQ face images dataset demonstrates that our approach outperforms existing methods. Besides this, our method presents two important advantages derived from the qualitative evaluation: (i) we obtain better protection results for lower noise levels than our competitors; and (ii) we protect the content from unauthorized use thereby protecting the individual's identity from potential misuse.

## 1 Introduction

Diffusion Models (11; 13; 34) are current state of the art for image generation, outperforming GANs in terms of image quality and mode coverage (4; 33). With the advent of diffusion models (25; 31; 24) we are able to achieve unprecedented accuracy and robustness towards generating high-fidelity images with enhanced model's ability to capture intricate details and complex patterns in data via textual prompts. These models have demonstrated significant performance across various applications including but not limited to image editing (9; 22), image-to-image translation (30), text-to-3D images synthesis (23; 41; 36), video generation (12; 1; 7), and anomaly detection in medical images (40). However, one application that has become extremely popular due to its versatility and ease of use is the customization of diffusion models with personal content (5; 28; 15). These models have the ability to create personalized content, by fine-tuning with just 3-5 user images a pre-trained diffusion model (e.g. Stable Diffusion) in order to learn to bind a unique new token to a novel concept. Textual Inversion (6), Dreambooth (29), and Custom Diffusion (16) are the seminal works in this direction. While these customization techniques are powerful tools to create user-specific content, they also pose substantial privacy risks since a malicious user could exploit these capabilities to generate deceptive images ('deep fakes'), which are visually indistinguishable from the real ones (21).

To prevent these privacy risks, there are currently some efforts that explore developing defense mechanisms, which inject a small imperceptible noise into images, that can protect against malicious use of customized diffusion models. Anti-customization methods aims to train T2I diffusion model with adversarial samples, generated with strong Projected Gradient Descent (PGD) (20). Pioneer work by (17) generates adversarial samples without modifying model's parameters. However, (38) fine-tuned all parameters of the DreamBooth model (28) when training to generate adversarial

---

*: Co-corresponding author

38th Conference on Neural Information Processing Systems (NeurIPS 2024).

samples. This is a highly inefficient method and therefore, inspired by Custom Diffusion (CD) (15), in CAAT (42) they propose to obtain adversarial samples only based on updating the cross-attention layers, since they undergo significant changes during fine-tuning. By freezing most of the model's parameters and only updating the key (K) and value (V) matrices, we can disrupt the connection between text and image during fine-tuning. This limits the model's ability to correctly classify images while still recognizing their content. Follow-up research focuses on further manipulating these cross-attention layers (18; 39)

In this paper, we propose a novel adversarial attack based method **Co**ncept **P**rotection by **S**elective **A**ttention **M**anipulation (*CoPSAM*) targeting only the cross-attention layers of the customized diffusion models for privacy protection. Different from (18), where the goal was to completely erase the class-specific token (e.g. 'face' or 'person'), in the current work the objective is to protect the user specific token against unauthorized use, while preserving individual's identity. Different from (42), here we aim to disrupt the similarity between user-specific and class-specific cross-attention maps. For this, we introduce the cosine similarity component into the loss function to aid the construction of the adversarial sample which helps to disrupt specifically the cross-attention maps of the fine-tuned diffusion model. The optimization process (i.e. minimization of the loss function) leverages the cosine similarity to create a divergence between the two maps, which effectively reduces their alignment. Our experiments show that this approach outperforms existing methods at the same low noise $\eta$-budget used in the PGD algorithm. To summarize, our contributions are the following:

- We propose *CoPSAM* for privacy protection in personalized diffusion models by considering only the cross-attention layers to strengthen the attack.

- A byproduct of our approach is that we are able to achieve better protection results with the adversarial attack at smaller budgets of noise, compared to competitive methods.

- Experimental validation on several men and women face images from the CelebA-HQ dataset (14) shows improved quantitative results outperforming the existing state-of-the-art methods.

## 2 Related Work

**Personalized T2I Diffusion Models.** Personalized T2I diffusion models have become very popular recently due to their versatility to produce user-tailored content. DreamBooth (28) was one of the first models to personalize content by fine-tuning the entire diffusion model, using 3-5 images to associate an unique identifier with a concept. Custom Diffusion (15) improves upon this by only fine-tuning the cross-attention layers, balancing personalization with broader model capabilities. SVDiff (8) offers another efficient approach by using Singular Value Decomposition for low-rank weight updates. In contrast, Textual Inversion (5) personalizes the content by learning new text embeddings for specific concepts, creating new tokens from a few images without altering the model itself.

**Privacy Protection in T2I Diffusion Models.** Recent research efforts have been devoted to propose several adversarial attack methods to protect personalized content from malicious manipulation. MIST (17) was one of the first methods, using imperceptible noise to disrupt a diffusion model's ability to replicate style and content without altering model parameters. Photoguard (32) employs encoder and diffusion attacks to defend against image editing operations. In contrast, Anti-DreamBooth (38) requires updating all model parameters for anti-personalization, which is very inefficient. Other methods (42) target cross-attention layers to disrupt text-image alignment, while some completely erase class-specific tokens, losing the identity in the generated images (18). Our method, *CoPSAM*, effectively safeguards image privacy while preserving individual identity.

## 3 Methodology

### 3.1 Preliminaries

**Latent Diffusion Models.** The Latent Diffusion Model (LDM) consists of two main components: (i) An autoencoder ($\mathcal{E}$) — that transforms an image $\mathcal{I}$ into a latent code $z_0 = \mathcal{E}(\mathcal{I})$ while the decoder ($\mathcal{D}$) reconstructs the latent code back to the original image such that $\mathcal{D}(\mathcal{E}(\mathcal{I})) \approx \mathcal{I}$; and (ii) Diffusion model — commonly a U-Net (27) based model — can be conditioned using class labels, segmentation

masks, or textual input. Let $\tau_\theta(y)$ represent the conditioning mechanism that converts a condition $y$ into a conditional vector for LDMs. The LDM model is refined using the noise reconstruction loss as demonstrated in eq. 1. We use Stable Diffusion v2.1 (35) as backbone model, which is based on Latent Diffusion Model (LDM) (26).

$$\mathcal{L}_{LDM} = \mathbb{E}_{z_0 \sim \mathcal{E}(x), y, \epsilon \sim \mathcal{N}(0,1)} \underbrace{\left( \|\epsilon - \epsilon_\theta(z_t, t, \tau_\theta(y))\|_2^2 \right)}_{\mathcal{L}_{rec}}. \tag{1}$$

where, $\epsilon_\theta$ is a conditional U-Net (27) that predicts the noise to be added. Particularly, T2I diffusion models aim to generate an image from a random noise $z_T$ given a conditioning prompt $\mathcal{P}$. For better readability, we denote text condition as $\mathcal{C} = \tau_\theta(\mathcal{P})$.

**Cross-attention Maps in T2I.** Cross-attention is a mechanism that aligns information between two different modalities. In the context of T2I diffusion models, these modalities are text prompt and image features, where these maps can be obtained from $\epsilon_\theta(z_t, t, \mathcal{C})$. More specifically, the cross-attention maps are derived from the deep features of the image which are projected into query matrix $Q_t = W_Q \cdot f_{z_t}$, and the textual embeddings — projected to key matrix $K = W_K \cdot \mathcal{C}$. The cross-attention maps are then accumulated as shown in eq. 2:

$$\mathcal{A}_t = softmax(Q_t \cdot K^T / \sqrt{d}) \cdot V \tag{2}$$

where, $d$ refers to latent dimension, and the cell $[\mathcal{A}_t]_{ij}$ defines the weight of $j$-th token on $i$-th token.

**Personalization in T2I.** Textual Inversion (6), DreamBooth (29), and Custom Diffusion (16) are seminal works in T2I personalization methods that aim to learn new concepts in the model given 3-5 images along with the text prompts, while retaining its prior knowledge. A common approach is to learn a novel text token through text encoding. The newly introduced token is initialized with a rarely occurring token embedding and optimized using reconstruction or customized loss functions during training to enhance the learning performance.

**Adversarial Attacks on T2I.** Adversarial attacks represent a sophisticated form of manipulation that introduces subtle, human-imperceptible perturbations to input data. These attacks primarily target deep learning classification models, which are renowned for their high accuracy. The goal is to induce imperceptible noise into the input data such that these models produce dramatically incorrect classification. An adversarial attack is formulated as follows: given an input $x$, obtain a perturbed input $x'$ such that :

$$\underset{x'}{argmax} \, \mathcal{L}_\theta(x') \quad s.t. \, \|x' - x\|_\infty \leq \eta \tag{3}$$

where, $\mathcal{L}_\theta$ is the loss function i.e. cross-entropy used to train the model with parameters $\theta$. We use the $\ell_\infty$-norm of noise budget given by $\eta$. Commonly, the strong PGD algorithm is used to construct the adversarial attack.

For T2I diffusion models, the main focus of this work, the above objective remains the same: obtaining an adversarial image $x'$ from a clean image $x$ perturbed with an imperceptible noise (within the given $\eta$-budget) such that when a T2I model is trained with the adversarial image, will lead it to generate images grossly different from a model trained with clean samples (and this can be easily detected by the human eye).

## 3.2 Concept Protection by Selective Attention Manipulation (*CoPSAM*)

As discussed in the literature (2; 9), cross-attention maps illustrate how text tokens influence image pixels, guiding T2I generation in diffusion models. Therefore, we start by analyzing the training process of Custom Diffusion through cross-attention maps. The whole pipeline of our approach *CoPSAM* is illustrated in Figure 1.

Given a textual guidance sequence $\Omega = \{\omega_1, \omega_2, ..., \omega_n\}$, we generate corresponding attention maps $\mathcal{A} = \{a_1, a_2, ..., a_n\}$, where $n$ is the number of words (text tokens). At timestep $t$, $a_t$ is derived from the forward diffusion process with latent code $z_t$ using eq. 2. To enhance privacy protection, our approach aims to disrupt this correlation by distancing the user-specific "<v>" token from the class-specific "<o>" token. The idea is that by pushing the "<v>" token away from the "<o>" token in the cross-attention maps (changing or reducing highlighted areas), the model loses its class-specific alignment. This results in the learned "<v>" token lacking both the necessary class-specific

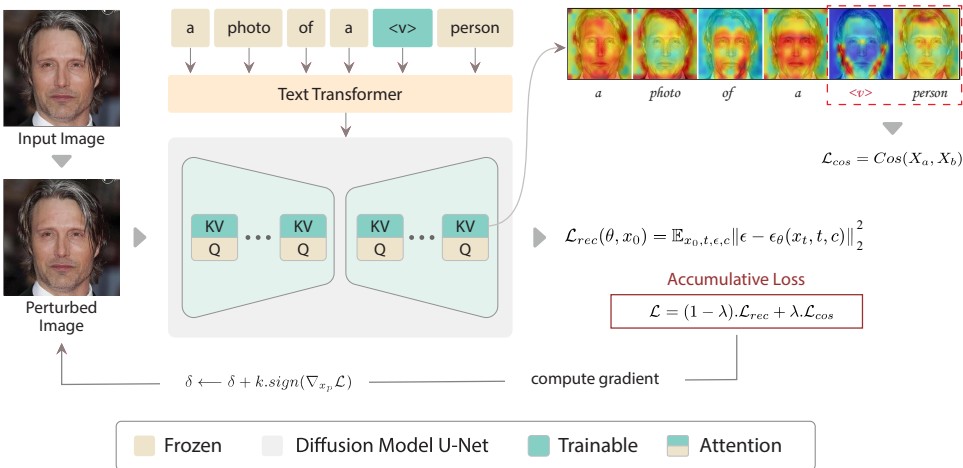

Figure 1: Illustration of our method *CoPSAM*.

information and the personalization information it should have acquired during the training process. Consequently, the generated images excludes identifiable information, ensuring greater privacy. This is achieved by minimizing the cosine similarity between "<v>" and "<o>" tokens. The optimal adversarial perturbation $\delta$ is obtained by applying the PGD algorithm with the overall loss function $\mathcal{L}$ (Eq 5). It consists of two components: (i) $\mathcal{L}_{rec}$ which is the standard reconstruction loss used to train the Latent Diffusion Model as in Eq. 1; and (ii) $\mathcal{L}_{cos}$ which is the cosine similarity loss given in Eq 4:

$$\mathcal{L}_{cos} = \frac{\mathcal{A}_v \cdot \mathcal{A}_o}{\|\mathcal{A}_v\|\|\mathcal{A}_o\|} \tag{4}$$

$$\mathcal{L} = (1 - \lambda) \cdot \mathcal{L}_{rec} + \lambda \cdot \mathcal{L}_{cos} \tag{5}$$

where, $\mathcal{A}_v$ and $\mathcal{A}_o$ are the corresponding cross attention maps of the user-specific token "<v>" and class-specific token "<o>", respectively and $\lambda$ is a blending parameter which controls the contribution of each of the two components in $\mathcal{L}$. The detailed algorithm is available in the Appendix.

Fig. 2 visualizes the result of *CoPSAM*'s impact on the respective cross-attention maps. Our customization prompt consists of a user-specific token "<v>" and a class-specific token "man/woman". In the cross-attention maps, red color highlights areas of the image which shows strong connection between text token and the generated image. Compared to the clean Custom Diffusion (first two rows), the impact of the specific token "<v>" on image generation significantly changes. In the unprotected models, the diffusion model captures the subject-specific token "<v>" (highlighted red areas on the face) and generates customized images. However, *CoPSAM* (last two rows) forces the model to avoid focusing on the key facial features when learning the "<v>" token, which were targeted by the clean Custom Diffusion model. For example, in the clean Custom Diffusion results, the main focus of "<v>" in the male cross-attention map is the mouth and jaw, while in the female cross-attention map, it is the mouth features. In contrast, with *CoPSAM*, the main focus in the male cross-attention map shifts to the nose and upper part of the ear, and in the female cross-attention map, it shifts to the top of the head. These findings confirm that *CoPSAM* effectively disrupt the image-text associations within the diffusion model, thereby protecting personal privacy.

## 4 Experimental Results

In this section, we quantitatively and qualitatively evaluate *CoPSAM* and compare it with existing methods in order to assess the performance of our approach for different noise budgets ($\eta$) and the impact of the newly introduced loss component, cosine similarity ($\mathcal{L}_{cos}$).

### 4.1 Experimental Setup

**Dataset.** To substantiate the relevance of *CoPSAM*, we used CelebA-HQ dataset (14), which is a high-resolution version of CelebA dataset (19), containing 10,177 unique celebrity identities and

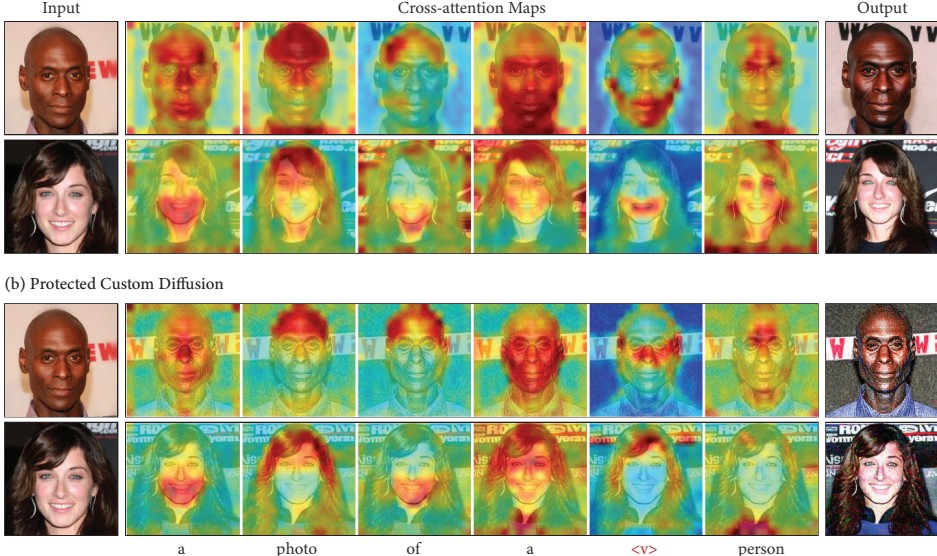

Figure 2: Illustration of cross-attention maps from clean Custom Diffusion (a) and *CoPSAM* (b). The significant difference on token "<v>" between our method and unprotected result indicates that our method effectively prevents model from focusing on the areas it should have been targeting.

202,599 face images. CelebA-HQ includes over 30,000 high-resolution $1024 \times 1024$ images of more than 1,000 celebrities. Our evaluation used a curated selection of 8 subjects from CelebA-HQ, ensuring a diverse representation across gender, ethnicity, and age. We reduced the image resolution to $512 \times 512$, which is the size required by the standard Stable Diffusion.

**Implementation details.** *CoPSAM*'s training was focused exclusively on the cross-attention layers of the Stable Diffusion model v2.1 (35), using a batch size of 2 and a learning rate of $1 \times 10^{-5}$ with 250 training steps. The conditioning prompt used was "a photo of a <v> man/woman.", where the <v> text token is initialized with '*ktn*'. For the PGD attack, we set $k = 2 \times 10^{-3}$ (step size) and a noise budget of $\eta = 8/255$ (unless stated otherwise) in $\ell_\infty$-norm. Regarding the blending parameter $\lambda$ used in the final loss function, we tried several values and found that the best results were obtained with $\lambda = 0.1$. All the experiments were run on a server with Nvidia A40 GPU.

**Comparison with SoTA.** We compared *CoPSAM* with existing attack methods aiming at T2I diffusion models, like Anti-DreamBooth (38), MIST (17), and CAAT (42). All these approaches represent the current state-of-the art and share a similar objective of protecting users' privacy.

**Evaluation metrics.** We apply standard metrics commonly used to evaluate the generated images in this study. Face Detection Failure Rate (FDFR), based on the RetinaFace detector (3), measures the absence of detectable faces, with higher values indicating more successful attacks. Identity Score Matching (ISM) calculates the cosine distance between face embeddings, where lower values signify better attack performance. Fréchet Inception Distance (FID) (10) measures the difference between generated and clean images, with higher values showing greater dissimilarity. SER-FIQ (37) evaluates facial image quality, with lower values representing more effective attacks.

## 4.2 Qualitative Evaluation

For qualitative evaluation, we first generate the adversarial samples using these models. Next, a clean T2I diffusion model is fine-tuned with these adversarial samples. Finally, we generate some inference images with the conditioning prompt: "a photo of a <v> man/woman".

We present a qualitative comparison in Figure 3. We can visually verify that our approach is able to generate more corrupted inference images compared to other methods with the same noise budget of $\eta = 8/255$. In this case, the existing methods fail to protect person's identity. That is because MIST used a simple image with a dense, repetitive, sharp boundary pattern to guide the added noise. This

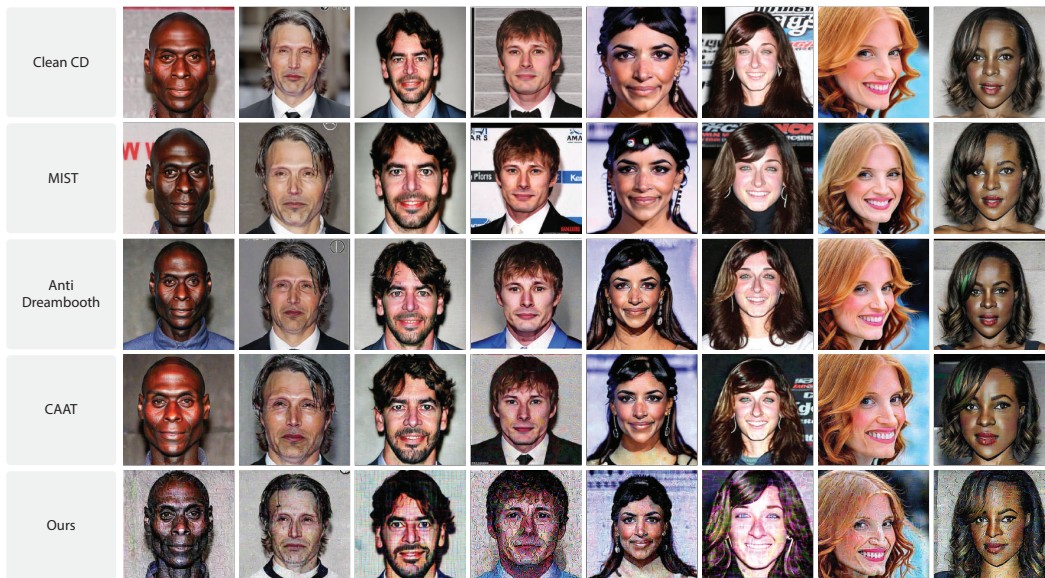

Figure 3: Comparison of images generated using different attack methods with the same noise budget of $\eta = 8/255$. From top to bottom, each group shows the results of the different methods. Clean CD refers to clean Custom Diffusion.

method is too simple and can only rely on the higher noise budget to produce an effective attack. Further, we observe that Anti-DreamBooth presents a low attack effectiveness due to its attempt to use the training process of clean images as a reverse guide for adding noise. Essentially, this approach attacks the global diffusion model, which is inefficient to break the semantic relation of the newly learned text token. While CAAT, targeting the cross-attention layers, is an effective and straightforward approach, in our results we observe that solely attacking the cross-attention layers based on $\mathcal{L}_{rec}$ is insufficient to be effective at lower noise budget like $\eta \leq 8/255$.

In our qualitative evaluation, we can visually observe that *CoPSAM* is able to obtain better perturbations at the same lower noise budget that is more effective in disrupting the customized T2I diffusion models. In other words, *CoPSAM* better protects personal data in comparison with other methods at lower budgets, by rendering it unusable for unauthorized purposes while simultaneously preserving the individual's identity.

Table 1: Effectiveness assessment using four evaluation metrics to compare different attack methods. *CoPSAM* achieved the best scores across all metrics, demonstrating its superior attack effectiveness.

| Attack method | ISM↓ | FDFR↑ | FID↑ | SER-FIQ↓ |
|---|---|---|---|---|
| No defence | 0.6607 | 0.0006 | 66.51 | 0.7396 |
| MIST | 0.6443 | 0.0006 | 74.75 | 0.7358 |
| Anti-DreamBooth | 0.6198 | 0.0012 | 87.03 | 0.7338 |
| CAAT | 0.5986 | 0.0018 | 96.06 | 0.7310 |
| Ours(*CoPSAM*) | **0.4993** | **0.0037** | **126.67** | **0.7279** |

## 4.3 Quantitative Evaluation

Table 1 presents the performance results of *CoPSAM* compared with other attack methods. For a fair comparison, we used original source codes and trained the models to generate adversarial samples with the same noise budget of $\eta = 8/255$. To obtain the results with each metric, we generated 200 images for each of the considered methods. The metric values indicate that the generated images are less similar to the original clean images in terms of quality, indicating that *CoPSAM* is highly effective at rendering it unusable for unauthorized purposes while preserving the individual's identity. *CoPSAM* achieves best scores across all the metrics.

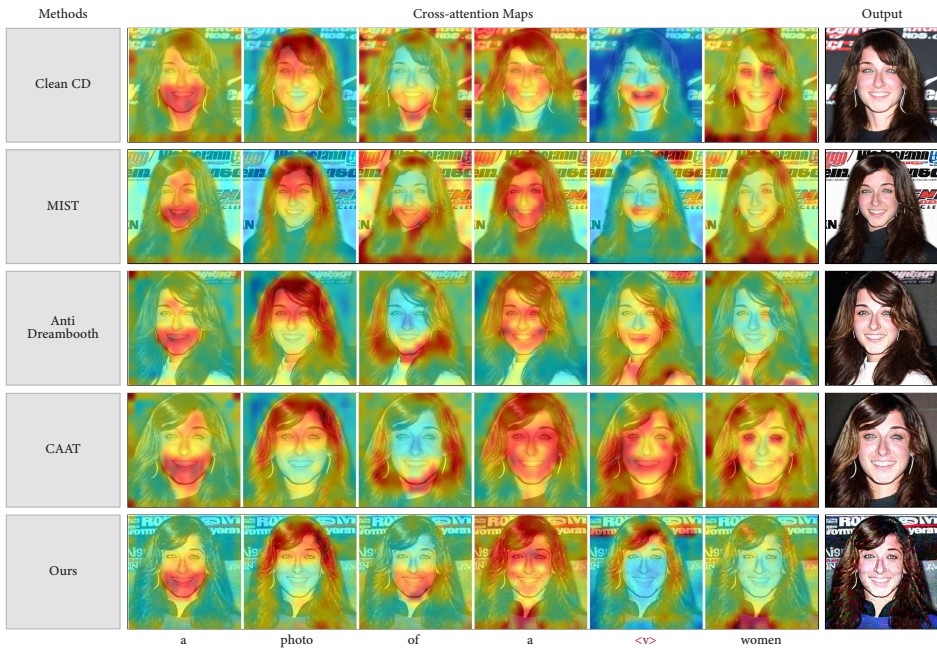

Figure 4: Visualizations of identity images, attention maps, and generated images before and after protection using various attack methods. Clean CD refers to clean Custom Diffusion.

At this point, we emphasize the specific usage of noise budget $\eta = 8/255 \approx 0.031$ as we observe this to be a good budget for maintaining the balance between imperceptible nature of the added noise and the protection offered for unauthorized usage. We also noticed that while the other methods may offer some privacy protection for this noise budget, the original results reported in their papers are obtained with a similar or higher budgets. For example, Anti-DreamBooth used $\eta = 0.05 \approx 12.75/255$, CAAT used $\eta = 0.1 \approx 25.5/255$, and MIST used $\eta = 0.03 \approx 8/255$. Thus, *CoPSAM* offers better privacy protection against unauthorized usage of personal data compared to the competitive methods.

### 4.4 Impact of Cosine Similarity

To evaluate the impact of the cosine similarity in the total loss function, in Figure 4 we visualize the attention maps and the corresponding generated images during the inference process. In general, the elements in red color indicates that the region receives more attention. In the first row of Figure 4, the red regions appear on the face for the token "<v>", indicating that the diffusion model pays more attention to the learned identity with "<v>". This is intuitive as the diffusion model progressively associates the token "<v>" more closely with the specific person. This observation persists for every method we compare to, which could be an undesirable effect for privacy protection. From the generated images, we observed that MIST and Anti-DreamBooth pay more attention on key facial features, indicating that these two methods have lesser effect on the diffusion model's identity learning. Alternatively, CAAT can somehow disperse attention across the global image, as it also targets the cross-attention layers, however, it still retains some focus on the correct identity region in the image. In contrast with the previous methods, *CoPSAM* successfully makes the "<v>" token significantly different from the clean image. Consequently, the diffusion model's guidance during generation is disrupted, preventing it from accurately recreating the original identity. This results in more indistinct images, enhancing privacy protection by making the learned token less recognizable and distinct from the original identity.

### 4.5 Impact of Noise Budget

The noise budget ($\eta$) refers to the maximum allowable perturbation that can be added to the input data to create the adversarial samples. An increased budget could result in perturbations that become more perceptible to the human eye. Figure 5 and Table 2 show the effectiveness of attack with

Table 2: Analysis across varying noise budgets $\eta$ for *CoPSAM* ("*" - indicates default budget).

| $\eta$-budget | ISM↓ | FDFR↑ | FID↑ | SER-FIQ↓ |
|---|---|---|---|---|
| $4/255$ | 0.6059 | 0.0006 | 85.06 | 0.7309 |
| $*8/255$ | 0.4993 | 0.0037 | 126.67 | 0.7279 |
| $16/255$ | 0.3623 | 0.0475 | 162.43 | 0.7206 |

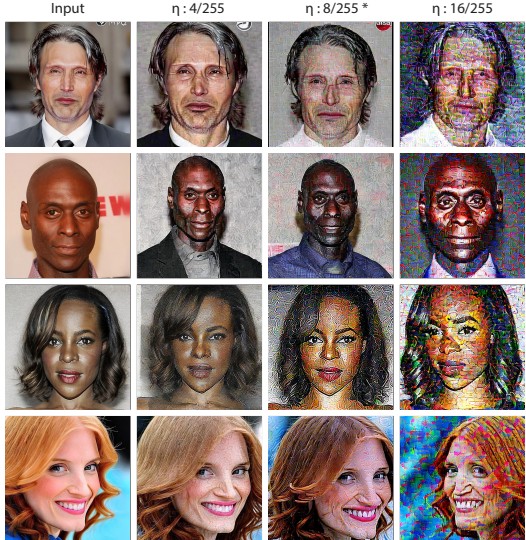

Figure 5: Visualizations across varying noise budgets for *CoPSAM* ("*" - indicates default budget).

different noise budgets. Specifically, with a noise budget of 4/255, the generated adversarial samples begin to show noticeable perturbations. At 8/255, the perturbations in the generated image are more pronounced, significantly showing visible changes in the image. At the highest noise budget of 16/255, the adversarial samples exhibit the largest change, making the generated images almost completely unrecognizable. This noise level ensures the highest level of privacy protection, as the images become highly blurred and indistinct. Remarkably, even with a small noise budget of 4/255, *CoPSAM* remains highly effective and is comparable to other methods operating at a higher noise budget of 8/255. This demonstrates the effectiveness and robustness of *CoPSAM*, achieving similar levels of privacy protection and adversarial effectiveness with substantially less noise.

## 5 Conclusion

In this paper, we proposed *CoPSAM*, a novel adversarial attack method that targets only the cross-attention layers of a T2I diffusion model. Our approach involves the careful construction of an imperceptible noise, which is added to clean samples to obtain the adversarial samples. This is achieved during the fine-tuning process by maximizing the discrepancy between the cross-attention maps corresponding to the user-specific token and class-specific token, respectively. The qualitative and quantitative validation, conducted on a subset of the CelebA-HQ face image dataset, demonstrates that *CoPSAM* outperforms existing methods. Therefore, our results highlight *CoPSAM*'s effectiveness in balancing strong privacy protection while simultaneously maintaining essential visual information.

## Acknowledgements

Xide Xu acknowledges the Chinese Scholarship Council (CSC) grant No.202306310064. This work is supported by Grants TED2021-132513B-I00 and PID2022-143257NB-I00 funded by MCIN/AEI/10.13039/501100011033, by the European Union NextGenerationEU/PRTR and by ERDF A Way of Making Europa, the Departament de Recerca i Universitats from Generalitat de Catalunya with reference 2021SGR01499, and the Generalitat de Catalunya CERCA Program.

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

## A  Algorithm

The *CoPSAM* algorithm is presented below.

---

**Algorithm 1** Algorithm for targeted cross-attention adversarial attack in T2I diffusion models

---

**Require:** Input image $x$, Latent Diffusion model $\mathcal{L}_{LDM}$, noise budget $\eta$, step size $k$, number of steps $N$. Tokens "<v>" (user-specific) and "<o>" (class-specific), $\mathcal{A}_{token}$ = cross attention map of token

**Require:** Perturbed Image $x_p$

1: Initialize adversarial perturbation $\delta \longleftarrow 0$, and protected image $x_p \longleftarrow x$
2: **for** t = 1 ... N **do**
3:     $\mathcal{L}_{cos} = Cos(\mathcal{A}_v, \mathcal{A}_o)$
4:     $\mathcal{L}_{rec} = \mathbb{E}_{z_0 \sim \mathcal{E}(x), y, \epsilon \sim \mathcal{N}(0,1)} \left( \|\epsilon - \epsilon_\theta(z_t, t, \tau_\theta(y))\|_2^2 \right)$
5:     $\mathcal{L} = (1 - \lambda) \cdot \mathcal{L}_{rec} + \lambda \cdot \mathcal{L}_{cos}$
6:     Update adversarial perturbation: $\delta \longleftarrow \delta + k \cdot sign(\nabla_{x_p} \mathcal{L})$
7:     $\delta \longleftarrow clip(\delta, -\eta, \eta)$
8:     Update the protected image: $x_p \longleftarrow x_p + \delta$
9: **end for**
10: **return** $x_p$

---

