# OpenReview forum: "Privacy Protection in Personalized Diffusion Models via Targeted Cross-Attention Adversarial Attack"
_NeurIPS.cc/2024/Workshop/SafeGenAi — SafeGenAi Poster_

### Official Review · Reviewer_ouLg · 2024-10-09
**The paper introduces improved privacy protection with smaller budgets of adversarial noise**

**Rating:** 6
**Confidence:** 3

**Review:**

**Strengths**

* The method introduces a cosine similarity loss between user-specific token and class-specific token to enhance privacy protection.
* The experimental results show that the proposed method can achieve better protection with smaller budgets of noise.
* The paper provides analyses and visualizations on impact of different protection components.

**Weaknesses**

* The idea of targetting on cross-attention layers has been repeatedly used in previous works.
* The experiments fail to include an essential competitive baseline (DisDiff), which also targetting on cross-attention, despite the paper mentioning it in the introduction.
* The annotations in Figure 1 do not match those in Section 3.2 of the main paper, e.g., "<o>" token and "Cos($X_a$, $X_b$)".

---

### Official Review · Reviewer_X8ei · 2024-10-09
**Insufficient Evaluation, Problems With Robustness**

**Rating:** 5
**Confidence:** 3

**Review:**

Summary: This paper builds upon the contribution of https://arxiv.org/pdf/2404.15081 by changing the loss function to be maximized via PGD into a convex combination of what it previously was and a new component. This new component is the cosine similarity between a class-specific token and an instance-specific token, as seen in https://arxiv.org/pdf/2208.12242, specific to whatever is going to be the likely target of a deepfake attack.

Pros: Some of the results of https://arxiv.org/pdf/2404.15081 are reproduced, and, to a certain extent, they are improved upon.

Cons: As papers like https://arxiv.org/pdf/2406.12027 discuss, this method's proper functioning hinges upon how the attack model is fine-tuned. For example, what if a different class-specific token is used? More importantly, it needs to be clarified which architecture the model was evaluated with after training on SD 2.1. It needs to be explained how comparable the metrics are between an approach that does not use class and user-specific tokens and an approach that does. Evaluation seems unclear; it should be done on other models.